# A Cognitive Framework for Learning Debiased and Interpretable Representations via Debiasing Global Workspace

**Jinyung Hong**[1]   **Eun Som Jeon**[2]   **Changhoon Kim**[1]   **Keun Hee Park**[1]

**Utkarsh Nath**[1]   **Yezhou Yang**[1]   **Pavan Turaga**[3]   **Theodore P. Pavlic**[1,4]

[1]**School of Computing and Augmented Intelligence**
[3]**School of Arts, Media, and Engineering**
[4]**School of Life Sciences**
Arizona State University, Tempe, AZ 85281, USA

[2]**Department of Computer Science and Engineering**
Seoul National University of Science and Technology, Seoul, 01811, Korea

## Abstract

When trained on biased datasets, Deep Neural Networks (DNNs) often make predictions based on attributes derived from features spuriously correlated with the target labels. This is especially problematic if these irrelevant features are easier for the model to learn than the truly relevant ones. Many existing approaches, called debiasing methods, have been proposed to address this issue, but they often require predefined bias labels and entail significantly increased computational complexity by incorporating extra auxiliary models. Instead, we provide an orthogonal perspective from the existing approaches, inspired by cognitive science, specifically Global Workspace Theory (GWT). Our method, *Debiasing Global Workspace* (DGW), is a novel debiasing framework that consists of specialized modules and a shared workspace, allowing for increased modularity and improved debiasing performance. Additionally, DGW enhances the transparency of decision-making processes by visualizing which features of the inputs the model focuses on during training and inference through attention masks. We begin by proposing an instantiation of GWT for the debiasing method. We then outline the implementation of each component within DGW. At the end, we validate our method across various biased datasets, proving its effectiveness in mitigating biases and improving model performance.

## 1 Introduction

Deep Neural Networks (DNNs) have achieved remarkable advancements across various domains, such as image classification (He et al., 2019; Xie et al., 2020), generation (Wang and Gupta, 2016; Kataoka et al., 2016), and segmentation (Luo et al., 2017; Zheng et al., 2014). However, DNNs often show limited generalization capability to out-of-distribution (OOD) data and are susceptible to biases present in their training datasets (Torralba and Efros, 2011). These biases occur when irrelevant features, like background color, correlate with target labels, causing the models to rely on these features for making predictions (Geirhos et al., 2020). This reliance on biased features leads to poor performance when the model encounters new data that does not share the same biases.

Biased datasets possess many *bias-aligned* samples, where irrelevant features correlate with the labels, and a small number of *bias-conflicting* samples, where these features do not align with the labels. Models trained on such data indeed tend to focus on the bias-aligned samples, leading to poor generalization (Hendrycks et al., 2021b,a).

Various debiasing methods have been proposed to prevent a network from relying on spurious correlations when trained on a biased dataset. Some methods assume that biased features are "easier" to learn than robust ones, leading to the use of auxiliary models that exploit these biased features to guide the main model's training (Nam et al., 2020; Sanh et al., 2020). Strategies such as re-weighting samples (Liu et al., 2021; Nam et al., 2020) and data augmentation (Kim et al., 2021; Lee et al., 2021) are common but often struggle with insufficiently diverse samples. Other approaches involve identifying specific biases before training (Hong and Yang, 2021; Kim et al., 2019; Li and Vasconcelos, 2019; Sagawa et al., 2019), allowing the model to ignore or correct these biases. Although effective, this requires accurate bias identification and extensive manual labeling (Bahng et al., 2020; Tartaglione et al., 2021).

In this work, we depart from the above perspectives and focus on a novel and completely different approach to implement a debiasing framework. In modern ML and AI, it has been argued that it is better to build an intelligent system from many interacting specialized modules rather than a single "monolithic" entity to deal with a broad spectrum of conditions and tasks (Goyal and Bengio, 2022; Minsky, 1988; Robbins, 2017). Toward this end, we focused on a cognitive science framework proposed to underlie perception, executive function, and consciousness: Global Workspace Theory (GWT). GWT is a crucial element of modern cognitive science that models human consciousness arising from integrating and broadcasting information across specialized, unconscious processes in the brain (Baars, 1993, 2005). Many recent studies proposing a deep-learning implementation of GWT (Bengio, 2017; Goyal et al., 2021; VanRullen and Kanai, 2021) have demonstrated their effectiveness in allowing a model to have general-purpose functionality, increased modularity, improved performance, and interpretable representation learning. This perspective is expected to be well suited for application in implementing debiasing methods.

Therefore, we propose the *Debiasing Global Workspace* (DGW), a novel instantiation of GWT for debiasing to eliminate the negative effect of the misleading correlations. Our debiasing approach involves specialized modules (acting as the specialists in GWT) and an attention-based information bottleneck (acting as the global workspace in GWT). This allows the model to achieve straightforward, functional modularity and effective debiasing performance while providing interpretable representation by visualizing which attributes are essential for accurate predictions and which are irrelevant and likely to cause errors.

The rest of this paper is organized as follows. We begin in Section 2 with a review of related work and relevant background literature. Then, in Section 3, we propose a conceptual modification of the GWT to implement a debiasing method. This involves defining specialized modules and the shared global workspace (Section 3.1). Then, we provide a step-by-step framework for defining the essential deep-learning components of our debiasing model within an AI system (Section 3.2). In Section 4, we empirically test our method on biased datasets, including Colored MNIST, Corrupted CIFAR10, and Biased FFHQ, and demonstrate that DGW effectively separates and understands intrinsic and biased features through both performance metrics and visualizations. Finally, we conclude with a discussion of future work and limitations of our approach in Section 5.

## 2 Related Work

There have been a variety of different debiasing methods for DNNs, and there also have been several connection points between GWT and neural networks. We review these approaches here.

### 2.1 Debiasing Methods

**Debiasing with predefined forms of bias or specific bias labels.** This method involves identifying specific biases before training (Hong and Yang, 2021; Kim et al., 2019; Li and Vasconcelos, 2019; Sagawa et al., 2019). The model then learns to ignore or correct these biases. Although effective, it depends on accurately identifying biases beforehand, which can be challenging. Another approach (Bahng et al., 2020; Tartaglione et al., 2021) uses bias labels to tag data, allowing the model

to differentiate between biased and unbiased data during training. This improves learning but requires extensive manual labeling.

**Debiasing using the easy-to-learn heuristic.** Biases are "easier" for models to learn (Nam et al., 2020) than intrinsic features. Techniques like dynamic training schemes, re-weighting samples, and data augmentation (Geirhos et al., 2018; Lee et al., 2021; Minderer et al., 2020; Li and Vasconcelos, 2019; Lim et al., 2023) help models focus on unbiased features. However, these methods struggle with insufficient diverse samples. Complex models can learn invariant features or correct representations but are difficult to design and train (Tu et al., 2022; Zhao et al., 2020; Agarwal et al., 2020; Bahng et al., 2020; Geirhos et al., 2018; Goel et al., 2020; Kim et al., 2019; Li et al., 2020; Minderer et al., 2020; Tartaglione et al., 2021; Wang et al., 2020).

**Others.** SelecMix (Hwang et al., 2022) creates new training samples by mixing pairs with similar labels but different biases, or different labels but similar biases, using an auxiliary contrastive model. Although effective, this adds significant training complexity. $\chi^2$ model (Zhang et al., 2023) learns debiased representations by identifying Intermediate Attribute Samples (IAS) and using a $\chi$-structured metric learning objective. However, its reliance on training dynamics to identify IASs makes it different from our approach and out of the scope of our study.

## 2.2 Deep Learning and Global Workspace Theory

In neuroscience and cognitive science, there is an ongoing effort to develop theories of consciousness (ToCs) to identify the neural correlates of consciousness, as reviewed by Seth and Bayne (2022). One such theory is the Global Workspace Theory (GWT) (Baars, 1993; Dehaene and Changeux, 2011; Mashour et al., 2020), which is inspired by the 'blackboard' architecture used in artificial intelligence. In this architecture, a centralized blackboard resource facilitates information sharing among specialized processors.

Recent studies have aimed to bridge the gap between neuroscience and deep learning, focusing on practical solutions for implementing a GWT using current deep learning components while considering the equivalent brain mechanisms (Goyal and Bengio, 2022; Minsky, 1988; Robbins, 2017; Goyal et al., 2021; Hong et al., 2024). Bengio (2017) emphasized learning high-level concepts by selecting key elements through attention, forming a low-dimensional conscious state similar to language, which aids in better representation learning. Mashour et al. (2020) details GWT's implementation in neuroscience, suggesting that consciousness arises from extensive information sharing across brain regions via a central network of neurons.

Inspired by GWT, our Debiasing Global Workspace (DGW) framework manages intrinsic and biased attributes in neural networks. DGW integrates information from intrinsic and bias specialists, ensuring disentangled representations are considered in decision making. Unlike prior works focusing on monolithic architecture or general-purpose learning, our approach uniquely applies these theories to debiasing neural networks.

## 3 Method

We propose the Debiasing Global Workspace (DGW), an instantiation of GWT for debiasing. DGW learns the composition of attributes in a dataset and provides interpretable explanations for the model's decisions. We introduce the conceptual framework of GWT for debiasing first (Section 3.1), its implementation in a deep learning framework next (Section 3.2), and the training objectives last (Section 3.3).

### 3.1 The Conceptual Instantiation of Debiasing Global Workspace

Figure 1 depicts a conceptual overview of our proposed DGW framework. The conceptual flow of the DGW proceeds through a sequence of steps that we describe in detail here.

**Step 0.** To learn disentangled representations of intrinsic and biased attributes, we introduce two specialists: intrinsic $\phi^i$ and biased $\phi^b$. In the original GWT, the specialists connect to the global workspace before any stimulus appears, coupling their latent spaces bidirectionally with the

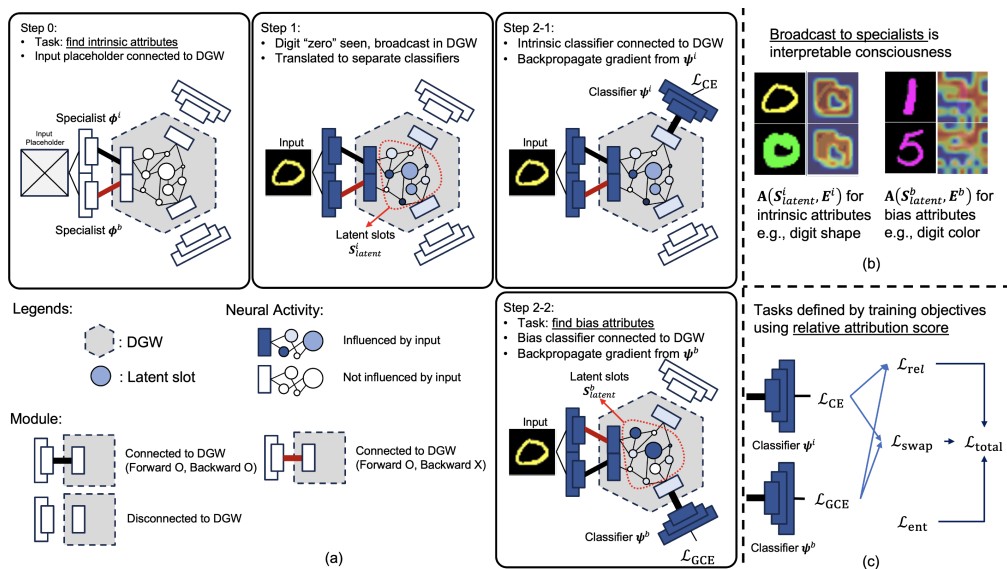

Figure 1: The conceptual framework of Debiasing Global Workspace (DGW). (a) Section 3.1: When attention selects inputs from specialists (Step 0), its latent space activation is copied into the DGW and immediately translated into representations suitable for each module (Step 1). We control which module is mobilized into the workspace to receive and process the corresponding data effectively. For example, upon recognizing the digit "zero," the corresponding classifiers are activated in the workspace. The classifier $\psi^i$ is initiated for intrinsic attributes (Step 2-1), and the classifier $\psi^b$ is activated for learning bias attributes (Step 2-2). (b) Broadcast in Section 3.2: The information broadcast in DGW can demonstrate interpretable representation for attribute learning. (c) Section 3.3: Unlike the original GWT, where task definitions can be preset in Step 0, we address them using our training objectives using relative attribution score. The generic figure is inspired by (VanRullen and Kanai, 2021, Fig. 3)

workspace. We modify this setup to control the connections to backpropagate different information to two specialists separately (black and red connections between specialists and DGW in Step 0 in Fig. 1). Specifically, the intrinsic and bias specialists function identically in the forward pass. However, during the backpropagation stage, only the intrinsic attribute encoder updates its parameters and learns, while the bias attribute encoder remains frozen and does not undergo parameter updates when the task of the model is to find intrinsic attributes.

**Step 1.** The DGW acts as an independent and intermediate shared latent space trained to perform unsupervised neural translation between the $C$ latent spaces from the specialized modules. The translation system is optimized to ensure that successive translation and back translation (e.g., a cycle from A to B, then back to A) return the original input (Goyal et al., 2021; VanRullen and Kanai, 2021). We implement specific operations to mimic the translation system by leveraging the residual operations (He et al., 2016) and a variant of mixup (Verma et al., 2019).

Posner (1994) argues that attention determines what information is consciously perceived and what is discarded in brains. In GWT, attention selects the information that enters the workspace. When a specific module is connected to the workspace through attention, its latent space activation vector is copied into the DGW. This internal copy serves as a bidirectional connection interface between the corresponding module and the DGW.

When a new stimulus, such as the digit "zero," appears, its latent activity transfers to the corresponding internal copy inside the workspace, initiating a broadcast to all other domains. This shared latent space ($\mathbf{S}_{latent}^i$ in Fig. 1) uses translations and back translations from all modules to compute and train via error backpropagation. We introduce a recurrent, top-down pathway, and it can sometimes be considered as a key to account for the global ignition property observed in the brain when an input reaches consciousness, and the corresponding module is mobilized into the conscious global workspace (VanRullen and Kanai, 2021).

**Step 2.** The incoming information is then immediately broadcast and translated (via the shared latent space) into the latent space of all other modules. In GWT, this translation process is automatic. However, we modify this to manually enforce learning of intrinsic and biased attribute representation with different loss functions. Specifically, we enforce the classifier $\phi^i$ to learn intrinsic attributes through error backpropagation from specific training objectives (Step 2-1 in Fig. 1). Step 2-2 simultaneously enforces the connection to the classifier $\psi^b$ and limits backpropagation to the intrinsic specialist $\phi^i$ to learn the bias attribute representations.

### 3.2 Roadmap to Implement Debiasing Global Workspace

Here, we present our deep-learning-based implementation of the DGW. It combines and organizes existing components for effective debiasing frameworks in a way that is consistent with the cognitive-science-inspired DGW framework described above.

**Two specialized modules and the shared workspace.** DGW uses two independent specialists, the intrinsic attribute encoder $\phi^i$ and the bias attribute encoder $\phi^b$. From these, we derive concatenated features $\mathbf{E} = [\phi^i(\mathbf{x}); \phi^b(\mathbf{x})] \in \mathbb{R}^{L \times D}$. To connect specialists and the shared workspace, we define $\mathbf{e} \in \{\mathbf{E}^i, \mathbf{E}^b\}$, where $\mathbf{E}^i = [\phi^i(\mathbf{x}); \mathrm{sg}(\phi^b(\mathbf{x}))]$ and $\mathbf{E}^b$ is vice versa, with $\mathrm{sg}(\cdot)$ as the stop-gradient operator. We introduce the Global Latent Attention (GLA) module, which acts as a shared workspace that encourages the synchronization among the input feature vector $\mathbf{E}$ via a latent feature representation $\mathbf{S}_{\text{latent}}$.

**Latent-slot binding specific to each input.** The GLA module uses a set number of latent embeddings or latent slots $C$. These latent slots represent the learnable embedding vectors in the DGW, and perform competitive attention (Vaswani et al., 2017) on the input features $\mathbf{e}$. We define $\mathbf{s}_{\text{latent}} \in \{\mathbf{S}^i_{\text{latent}}, \mathbf{S}^b_{\text{latent}}\} \in \mathbb{R}^{C \times D}$ where $C^i$ is number of slots for intrinsic features and $C^b$ for bias features, with $C = C^i + C^b$. The attention mechanism is such that:

$$\mathbf{A}(\mathbf{e}, \mathbf{s}_{\text{latent}}) = \mathrm{softmax}\left(\frac{k(\mathbf{e}) \cdot q(\mathbf{s}_{\text{latent}})^\top}{\sqrt{D}}\right) \in \mathbb{R}^{C \times L}, \tag{1}$$

where, $k, q$ are linear projection matrices, and the softmax function normalizes the slots, creating competition among them. The slots are refined iteratively using the following:

$$\mathbf{s}^{(n+1)}_{\text{latent}} = \mathrm{GRU}\left(\mathbf{s}^{(n)}_{\text{latent}}, \mathrm{Normalize}\left(\mathbf{A}(\mathbf{e}, \mathbf{s}^{(n)}_{\text{latent}})^\top\right) \cdot v(\mathbf{E})\right), \tag{2}$$

where, $\mathbf{s}^{(n)}_{\text{latent}}$ is the latent slot representation after $n$ iterations, GRU (Cho et al., 2014) is a recurrent neural network, and $v$ is another liner projection matrix. The initial slots $\mathbf{s}^{(0)}_{\text{latent}}$ are initialized with learnable queries following (Jia et al., 2022).

The above computations can be considered to implement a shared global workspace (Goyal et al., 2021; Hong et al., 2024) as they enable different parts of the model to compete for attention, integrating and broadcasting information similar to GWT.

**Broadcast updated information to specialists.** Specialists update their states using information from the shared workspace. The inverted cross-attention mechanism allows specialists to query and interact with updated latent slots $\mathbf{s}^{(n+1)}_{\text{latent}}$, updating their states through:

$$\bar{\mathbf{e}} = \mathbf{e} \oplus \left(\mathbf{A}\left(\mathbf{s}^{(n+1)}_{\text{latent}}, \mathbf{e}\right) \cdot v\left(\mathbf{s}^{(n+1)}_{\text{latent}}\right)\right) \in \mathbb{R}^{L \times D}, \tag{3}$$

where $v$ is a linear projection matrix. Here, as the meaning of information broadcast, $\oplus$ can be instantiated with various computational operations, including a residual connection (He et al., 2016). The other way of operation is a modified version of Manifold Mixup (Verma et al., 2019), which interpolates feature embeddings to capture higher-level information:

$$\bar{\mathbf{e}} = \mathrm{Mix}_\alpha\left(\mathbf{e}, \left(\mathbf{A}\left(\mathbf{s}^{(n+1)}_{\text{latent}}, \mathbf{e}\right) \cdot v\left(\mathbf{s}^{(n+1)}_{\text{latent}}\right)\right)\right),$$

where $\mathrm{Mix}_\alpha(a, b) = \alpha \cdot a + (1 - \alpha) \cdot b$, and $\alpha \sim \mathrm{Beta}(\beta, \beta)$. The updated feature vector $\bar{\mathbf{e}}$ is then fed to the classifier $\psi^i$ and $\psi^b$. We compare the performance of using residual connections versus our modified Manifold Mixup in Section 4.1.

In GWT, the information broadcast through the global workspace is a necessary and sufficient condition for conscious perception (VanRullen and Kanai, 2021). Intuitively, the attention mask $\mathbf{A}(\mathbf{s}_{\text{latent}}^{(n+1)}, \mathbf{e})$ can be seen as artificial phenomenal consciousness, indicating the immediate subjective experience of sensations and perceptions. These non-negative relevance scores depend on $\mathbf{x}$ through the averaged attention weight, allowing us to show interpretable representations for intrinsic and biased attributes in our analysis (Section 4.2).

### 3.3 Training Objectives

Here, we summarize the objective functions to train our framework. We have two linear classifiers $\boldsymbol{\psi}^i$ and $\boldsymbol{\psi}^b$ that take the updated concatenated vector $\bar{e}$ from the previous module as input to predict the target label $y$. Our training objectives consist of: i) the relative attribute score learning phase, and ii) the attribute composition phase.

**Relative attribute score learning phase.**    In this phase, we define two tasks within the conceptual framework for: identifying intrinsic attributes and identifying biased attributes. Without specific information about bias types, we utilize the relative difficulty score of each data sample, as proposed by Nam et al. (2020). Specifically, we train $\phi^b$, $\mathbf{S}_{\text{latent}}^b$, and $\boldsymbol{\psi}^b$ to focus on bias attributes using generalized cross entropy (GCE) (Zhang and Sabuncu, 2018), while $\phi^i$, $\mathbf{S}_{\text{latent}}^i$ and $\boldsymbol{\psi}^i$ are trained with the cross entropy (CE) loss. Samples with high CE loss from $\boldsymbol{\psi}^b$ are considered bias-conflicting compared to those with low CE loss. We define the relevance score function:

$$\text{Score}(\bar{e}, y) \triangleq CE(\boldsymbol{\psi}^b(\bar{e}), y) \Big/ \Big( CE(\boldsymbol{\psi}^i(\bar{e}), y) + CE(\boldsymbol{\psi}^b(\bar{e}), y) \Big).$$

Thus, the objective function is defined using the above relative difficulty score of each data sample:

$$\mathcal{L}_{\text{rel}} = \text{Score}(\bar{e}, y) \cdot CE(\boldsymbol{\psi}^i(\bar{e}), y) + \lambda_{\text{rel}} GCE(\boldsymbol{\psi}^b(\bar{e}), y),$$

where $\lambda_{\text{rel}}$ is the weight that adjusts between two loss terms. This loss function balances the learning between intrinsic and biased attributes, ensuring effective identification and separation of these attributes during the training phase.

**Attribute-composition phase.**    We swap the disentangled latent vectors among the training sets (Lee et al., 2021). We randomly permute the intrinsic and bias features in each mini-batch, creating $\mathbf{E}_{\text{swap}} = [\phi^i(\mathbf{x}); \phi_{\text{swap}}^b(\mathbf{x})]$ where $\phi_{\text{swap}}^b(\mathbf{x})$ denotes the randomly permuted bias attributes. This process produces augmented bias-conflicting latent vectors. As similar as the definition of $\mathbf{e}$, we define $\mathbf{e}_{\text{swap}} \in \{\mathbf{E}_{\text{swap}}^i, \mathbf{E}_{\text{swap}}^b\}$ and generate $\bar{\mathbf{e}}_{\text{swap}}$ following the same process described in eqs 1, 2 and 3. The objective function for this phase is:

$$\mathcal{L}_{\text{swap}} = \text{Score}(\bar{e}, y) \cdot CE(\boldsymbol{\psi}^i(\bar{e}_{\text{swap}}), y) + \lambda_{\text{swap}} GCE(\boldsymbol{\psi}^b(\bar{e}_{\text{swap}}), \tilde{y}),$$

where $\tilde{y}$ denotes target labels for permuted bias attributes $\phi_{\text{swap}}^b(\mathbf{x})$, and $\lambda_{\text{swap}}$ is the balancing weight between two loss terms. Notice that the relevance score $\text{Score}(\bar{e}, y)$ is re-used to reduce computational complexity. This loss function implies that by swapping bias features, the model learns to handle a wider variety of bias-conflicting samples, improving its ability to generalize beyond the specific biases present in the training data. Consequently, the model becomes more robust as it learns to focus on intrinsic features while disregarding spurious correlations, resulting in better performance on unbiased data. Furthermore, augmenting the training data in this manner helps the model generalize better to new, unseen data by exposing it to a wider range of possible biases during training.

**Entropy regularization.**    We empirically incorporate an additional regularization term on the latent slot attention mask to enhance performance:

$$\mathcal{L}_{\text{ent}} = H(\mathbf{A}(\mathbf{s}_{\text{latent}}^{(n)}, \mathbf{e})) + H(\mathbf{A}(\mathbf{s}_{\text{latent}}^{(n)}, \mathbf{e}_{\text{swap}})),$$

where $\mathbf{A}(\mathbf{s}_{\text{latent}}^{(n)}, \mathbf{e})$ and $\mathbf{A}(\mathbf{s}_{\text{latent}}^{(n)}, \mathbf{e}_{\text{swap}})$ are attention masks from the last iteration of eq. 2. Minimizing entropy $H(\mathbf{A}) = H(a_1, \ldots, a_{|\mathbf{A}|}) = (1/|\mathbf{A}|) \sum_i -a_i \cdot \log(a_i)$ encourages the attention masks to be consistent over the input features captured by the latent slots. This regularization ensures the model's attention remains focused and interpretable across different input scenarios.

**Final loss.** The total loss function is a combination of the above components: $\mathcal{L}_{total} = \mathcal{L}_{rel} + \lambda_{swap} \cdot \mathcal{L}_{swap} + \lambda_{ent} \cdot \mathcal{L}_{ent}$. Here, $\lambda_{swap}$ and $\lambda_{ent}$ are weights that adjust the importance of the feature augmentation and entropy regularization, respectively. This comprehensive loss function ensures balanced training that enhances the model's ability to learn and generalize effectively while maintaining interpretability and robustness.

## 4 Experiments

Here, we present our experimental results, focusing on performance evaluation on various biased datasets (Section 4.1), interpretable analysis for attribute-centric representation learning (Section 4.2), and additional qualitative and quantitative analyses (Section 4.3).

**Datasets.** Following the previous work (Lee et al., 2021), we used three well-known benchmark datasets for debiasing methods to evaluate DGW's performance and interpretability:

- **Colored MNIST (C-MNIST)** and **Corrupted CIFAR10 (C-CIFAR-10)**: These synthetic datasets are designed to test model generalization on unbiased test sets by varying the ratio of bias-conflicting samples (0.5%, 1%, 2%, and 5%).
- **Bias FFQH (BFFHQ)**: This real-world dataset from FFHQ (Karras et al., 2019) contains face images annotated with age (intrinsic attribute) and gender (bias attribute). Most samples are young women and old men, creating a high correlation between age and gender. For BFFHQ, we included 0.5% bias-conflicting samples in the training set and used a bias-conflicting test set to ensure robust evaluation.

At inference time, we evaluate the models on clean data where no bias-conflicting samples exist.

### 4.1 Performance Evaluation

**Baselines.** Our set of debiasing baselines includes six different approaches[1]: Vanilla network, HEX (Wang et al., 2018), EnD (Tartaglione et al., 2021), ReBias (Bahng et al., 2020), LfF (Nam et al., 2020), and LFA (Lee et al., 2021). Vanilla refers to the classification model trained only with the original cross-entropy (CE) loss without debiasing strategies. EnD leverages the explicit bias labels, such as the color labels in the C-MNIST dataset, during the training phase. HEX and ReBias assume an image's texture as a bias type, whereas LfF, LFA, and our method do not require any prior knowledge about the bias type. Furthermore, we configure a naive debiasing approach integrated with GWT implementation: V+CCT. CCT (Hong et al., 2024) proposed an instantiation of GWT applicable to implement an interpretable model. To compare our DGW, we simply configure the direction fusion of the Vanilla network with CCT as a GWT debiasing method.

**Implementation details.** Following the implementation details from Lee et al. (2021), we used a fully connected network for attribute encoders with three hidden layers for C-MNIST and ResNet-18 for C-CIFAR-10 and BFFHQ. We employed a fully connected classifier with double the hidden units to handle the combined output from the intrinsic attribute encoder $\phi^i$ and the bias attribute encoder $\phi^b$.

During testing, only the intrinsic classifier $\psi^i(\mathbf{e})$ was used for final predictions. We used batch sizes of 256 for C-MNIST and C-CIFAR-10, and 64 for BFFHQ, respectively. Two concepts and size of 8 were used for C-MNIST, 5 and 16 for C-CIFAR-10, and 10 and 32 for BFFHQ, respectively. We trained our model and baselines with three trials and reported the averaged accuracy and standard deviation. More details of experimental settings are explained in the Appendix C.4.

**Performance Comparison.** Table 1 shows that ReBias outperforms DGW on C-MNIST because it uses additional predefined bias labels. This gives ReBias a specific advantage. However, DGW excels without needing predefined bias labels, making it more versatile. DGW, with all operators,

---

[1]We only establish baselines that can be directly tested. For example, $\chi^2$ (Zhang et al., 2023) is not included because its code is not publicly available, and SelecMix (Hwang et al., 2022) is not included because it is a data-augmentation method that differs from our method category and has high training complexity, taking over approximately four times longer than our method. Additionally, although the authors of SelecMix claim it runs

Table 1: Test accuracy (%) on unbiased test sets of C-MNIST and C-CIFAR-10, and the bias-conflicting test set of BFFHQ with varying ratio of bias-conflicting samples. (∗) denotes methods tailored to predefined forms of bias, (°) methods using bias labels, (†) methods relying on the easy-to-learn heuristic, and (‡) methods combined with GWT. V+CCT indicates the direct integration of Vanilla and CCT. DGW+M refers to DGW with our mixup strategy, and DGW+R refers to DGW with residual connection. Performance for HEX and EnD is from (Lee et al., 2021), while results for Vanilla, ReBias, LfF, LFA, V+CCT and DGW are from our evaluation. The best-performing results are shown in bold, and the second-best results are underlined.

| Dataset | Ratio (%) | Vanilla | HEX* | EnD° | ReBias* | LfF† | LFA† | V+CCT‡ | DGW+M‡ | DGW+R‡ |
|---|---|---|---|---|---|---|---|---|---|---|
| C-MNIST | 0.5 | $36.2_{\pm1.8}$ | $30.3_{\pm0.8}$ | $34.3_{\pm1.2}$ | $\mathbf{72.2}_{\pm1.5}$ | $47.5_{\pm3.0}$ | $67.4_{\pm1.7}$ | $26.3_{\pm1.1}$ | $68.9_{\pm2.8}$ | $\underline{70.3}_{\pm1.2}$ |
| | 1.0 | $50.8_{\pm2.3}$ | $43.7_{\pm5.5}$ | $49.5_{\pm2.5}$ | $\mathbf{86.6}_{\pm0.6}$ | $64.6_{\pm2.5}$ | $79.0_{\pm1.0}$ | $40.1_{\pm2.1}$ | $\underline{81.3}_{\pm1.2}$ | $77.4_{\pm0.4}$ |
| | 2.0 | $65.2_{\pm2.1}$ | $56.9_{\pm2.6}$ | $68.5_{\pm2.2}$ | $\mathbf{92.7}_{\pm0.3}$ | $74.9_{\pm3.7}$ | $85.0_{\pm0.8}$ | $56.2_{\pm1.8}$ | $84.6_{\pm1.5}$ | $\underline{85.3}_{\pm0.7}$ |
| | 5.0 | $81.6_{\pm0.6}$ | $74.6_{\pm3.2}$ | $81.2_{\pm1.4}$ | $\mathbf{97.1}_{\pm0.6}$ | $80.2_{\pm0.9}$ | $88.7_{\pm1.3}$ | $73.4_{\pm0.8}$ | $88.9_{\pm0.2}$ | $\underline{89.1}_{\pm0.6}$ |
| C-CIFAR-10 | 0.5 | $22.8_{\pm0.3}$ | $13.9_{\pm0.1}$ | $22.9_{\pm0.3}$ | $20.8_{\pm0.2}$ | $25.0_{\pm1.5}$ | $27.9_{\pm1.0}$ | $15.2_{\pm0.3}$ | $\underline{29.6}_{\pm0.5}$ | $\mathbf{30.4}_{\pm2.2}$ |
| | 1.0 | $26.2_{\pm0.5}$ | $14.8_{\pm0.4}$ | $25.5_{\pm0.4}$ | $24.4_{\pm0.4}$ | $31.0_{\pm0.4}$ | $34.3_{\pm0.6}$ | $20.6_{\pm0.4}$ | $\mathbf{34.9}_{\pm0.4}$ | $\underline{33.6}_{\pm2.4}$ |
| | 2.0 | $31.1_{\pm0.6}$ | $15.2_{\pm0.5}$ | $31.3_{\pm0.4}$ | $29.6_{\pm2.9}$ | $38.3_{\pm0.4}$ | $40.3_{\pm2.4}$ | $24.6_{\pm0.5}$ | $\underline{41.3}_{\pm1.0}$ | $\mathbf{42.0}_{\pm1.9}$ |
| | 5.0 | $42.0_{\pm0.3}$ | $16.0_{\pm0.6}$ | $40.3_{\pm0.9}$ | $41.1_{\pm0.2}$ | $48.8_{\pm0.9}$ | $\underline{50.3}_{\pm1.1}$ | $35.6_{\pm0.8}$ | $\mathbf{52.3}_{\pm0.8}$ | $\underline{50.3}_{\pm1.9}$ |
| BFFHQ | 0.5 | $54.5_{\pm0.6}$ | $52.8_{\pm0.9}$ | $56.9_{\pm1.4}$ | $58.0_{\pm0.2}$ | $63.6_{\pm2.9}$ | $59.5_{\pm3.8}$ | $52.6_{\pm1.1}$ | $\mathbf{66.9}_{\pm1.0}$ | $\underline{65.6}_{\pm3.3}$ |

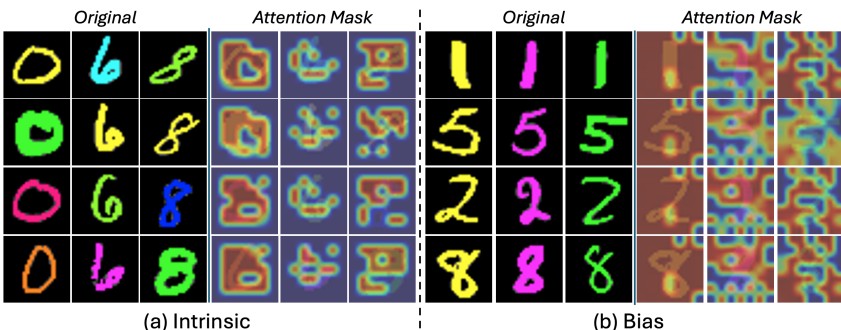

Figure 2: Visualization of $\mathbf{A}^i$ and $\mathbf{A}^b$ for the C-MNIST dataset

also outperforms LFA across all datasets, demonstrating its robustness and flexibility in debiasing image classification tasks. Furthermore, the poor performance of V+CCT highlights the importance of finding the proper configuration for debiasing methods, indicating the effectiveness of our DGW configuration as a debiasing method.

## 4.2 Analysis for Interpretable Attribute Representation

To make the analysis of interpretable attribute representation learning in our model more intuitive, let us explore the attention mask patterns $\mathbf{A}(\mathbf{s}_{\text{latent}}^{(n+1)}, \mathbf{e})$ for the C-MNIST and C-CIFAR-10 datasets. In the broadcast in our formulation ($\mathbf{A}(\mathbf{s}_{\text{latent}}^{(n+1)}, \mathbf{e})$ in eq. 3)), DGW generates two attention masks: $\mathbf{A}^i = \mathbf{A}(\mathbf{S}_{\text{latent}}^i, \mathbf{E}^i)$ for intrinsic attributes, focusing on essential features like shape, and $\mathbf{A}^b = \mathbf{A}(\mathbf{S}_{\text{latent}}^b, \mathbf{E}^b)$ for biased attributes, capturing non-essential features like color.

For the C-MNIST dataset, intrinsic attention masks highlight the shapes of the digits, ignoring colors. For instance, the digits "0," "6," and "8" consistently highlight shape regions (Fig. 2(a)), showing that the model focuses on shape for classification. Conversely, bias attention masks highlight color regions, not shapes. Digits "1," "5," "2," and "8" in yellow/magenta/green show nearly identical masks (Fig. 2(b)), indicating a focus on color. This confirms that the biased components of DGW capture color information, which is irrelevant for digit recognition.

For the C-CIFAR-10 dataset, intrinsic masks focus on uncorrupted parts of the images (Fig. 3(b)), highlighting true object features. For example, masks for a truck, car, dog, and horse highlight uncorrupted areas, avoiding noise. Bias masks, on the other hand, focus on corrupted areas, showing

on an RTX 3090, we found that our environment with a 24GB RTX A6000 could not handle the real-life dataset BFFHQ, indicating significant computational resource requirements.

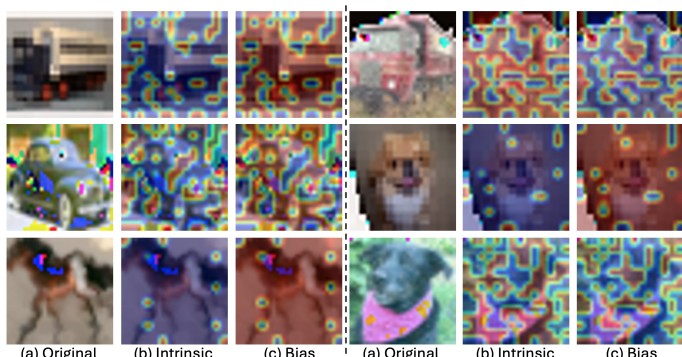

Figure 3: Visualization of $\mathbf{A}^i$ and $\mathbf{A}^b$ for the C-CIFAR10 dataset

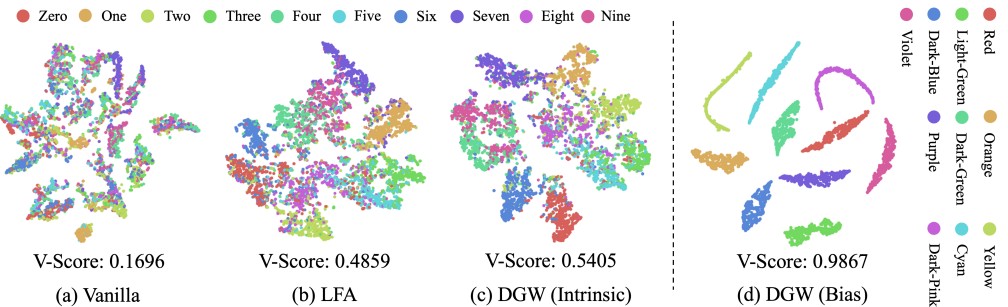

Figure 4: t-SNE plots for intrinsic and bias features on C-MNIST (with 0.5% setting).

no overlap with intrinsic masks (Fig. 3(c)). This complementary relationship illustrates the effective segregation of essential (intrinsic) and non-essential (biased) information.

In summary, for C-MNIST, intrinsic masks focus on digit shapes, while bias masks focus on colors. For C-CIFAR-10, intrinsic masks highlight uncorrupted parts, and bias masks cover corrupted parts. This clear separation supports the model's robustness and interpretability, ensuring decisions are based on relevant features while ignoring spurious correlations. More visualization results can be found in Appendix C.5.

### 4.3 Quantitative and Qualitative Analysis

We provide additional analysis to compare our DGW (DGW+M in Table 1) method with Vanilla and LFA (Lee et al., 2021). More experimental results with different settings can be found in Appendix C.6.

**t-SNE and Clustering.** We measure clustering performance using t-SNE (van der Maaten and Hinton, 2008) and V-Score (Rosenberg and Hirschberg, 2007) on features from various models capturing intrinsic and bias attributes on C-MNIST. V-Score represents homogeneity and completeness, with higher values indicating better clustering. In Fig. 4, our DGW's $\phi^i$ captures intrinsic attributes effectively, resulting in tighter clusters and better separation, as indicated by the V-Score. Bias attributes are well captured by the $\phi^b$, as shown in Fig. 4(d).

**Model Similarity.** We visualize model similarity using Centered Kernel Alignment (CKA) (Raghu et al., 2021; Kornblith et al., 2019; Cortes et al., 2012), comparing similarities between all pairs of layers for different models. In this analysis, I and B denote $\phi^i$ and $\phi^b$. As shown in Fig. 5, Vanilla and LFA possess similar weights across many layers, while DGW shows fewer similarities in both initial and deeper layers, indicating different behavior across layers compared to baselines.

**Model Reliability.** We evaluate model generalizability using Expected Calibration Error (ECE) and Negative Log Likelihood (NLL) (Guo et al., 2017). ECE measures calibration error, and NLL

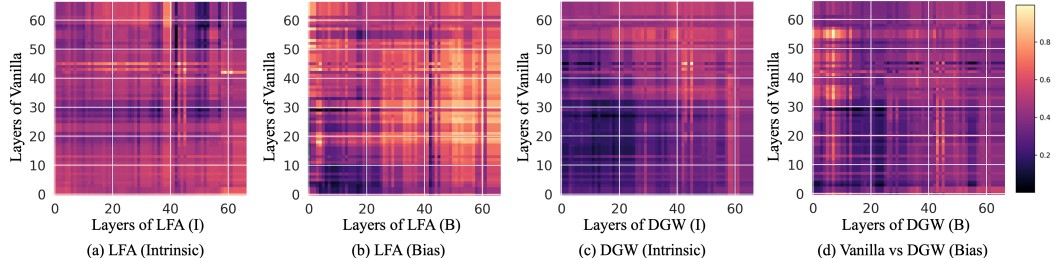

Figure 5: Representations of similarities for vanilla and different methods with all pairs of layers on C-CIFAR-10 (0.5% setting). High similarity score denotes high values.

Table 2: ECE (%) and NLL under different settings on C-CIFAR-10.

| Ratio (%): | 0.5 | | 1.0 | | 2.0 | | 5.0 | |
|---|---|---|---|---|---|---|---|---|
| Model | ECE | NLL | ECE | NLL | ECE | NLL | ECE | NLL |
| Vanilla | 13.75 | 5.99 | 13.14 | 9.87 | 12.25 | 6.65 | 13.76 | 5.99 |
| LFA | 12.09 | 5.81 | **11.45** | 7.27 | 10.25 | 5.14 | 7.56 | 3.09 |
| DGW (Ours) | **11.85** | **5.71** | 11.53 | **6.88** | **9.96** | **4.41** | **7.55** | **3.01** |

assesses probabilistic quality. As shown in Table 2, DGW consistently has the lowest ECE and NLL, indicating better generalizability compared to baselines.

## 5   Conclusion

In this work, we introduced Debiasing Global Workspace (DGW), a framework designed to learn debiased representations of attributes in neural networks. By leveraging attention mechanisms inspired by the Global Workspace Theory, our method effectively differentiates between intrinsic and biased attributes, enhancing both performance and interpretability. Comprehensive evaluations across various biased datasets demonstrated that DGW improves model robustness and generalizability on biased data and provides interpretable insights into the model's decision-making process. Our approach results in tighter clusters and better model separation, indicating superior performance in both intra- and inter-classification tasks. Furthermore, DGW shows enhanced model reliability and generalizability, making it a better solution for addressing biases in real-world applications. Future work could focus on reducing this complexity, exploring the scalability of DGW to even larger and more diverse datasets, and extending the framework into a general-purpose drop-in layer to enhance robust performance across a wider range of image recognition tasks.

**Limitations.**   We acknowledge that introducing our modules can increase training complexity, including model size and training time. This represents a trade-off between performance and decision-making transparency. Although our additional overhead is minimal, further analysis is necessary to optimize and streamline the process.

## Acknowledgments and Disclosure of Funding

This work was supported by NSF EFMA-2223839.

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

# Appendix

## A    Reproducibility

All source codes, figures, models, etc., are available at `https://github.com/jyhong0304/debiasing_global_workspace`.

## B    Background

**Object-Centric Representation Learning.**    Humans outperform sophisticated AI technologies due to our exceptional ability to recombine previously acquired knowledge, allowing us to extrapolate to novel scenarios (Fodor and Pylyshyn, 1988; Goyal and Bengio, 2022; Greff et al., 2020). Pursuing representations that generalize compositionally has been a significant research topic, with object-centric representation learning (Burgess et al., 2019; Greff et al., 2019; Locatello et al., 2020; Chang et al., 2022; Jia et al., 2022) emerging as a prominent effort. This approach represents each object in an image with a unique subset of the image's latent code, enabling compositional generalization due to its modular structure.

Due to its simple yet effective design, Slot-Attention (SA) (Locatello et al., 2020) has gained significant attention in unsupervised object-centric representation learning. Its iterative attention mechanism allows SA to learn and compete between slots for explaining parts of the input, showing a soft clustering effect on visual inputs (Locatello et al., 2020). Some recent works on implementing a cognitive architecture using object-centric methods have been proposed (Hong et al., 2024; Didolkar et al., 2023). Our approach also emphasizes compositional generalization in debiasing learning, using the slot-based method to implement a crucial module. The benefits of this method are noteworthy and deserve further exploration.

## C    Further Experimental Results and Details

In this section, we explain further experimental results and details. All experiments are conducted with three different random seeds and $95\%$ confidence intervals.

### C.1    Hardware Specification of The Server

The hardware specification of the server that we used to experiment is as follows:

- CPU: Intel® CoreTM i7-6950X CPU @ 3.00GHz (up to 3.50 GHz)
- RAM: 128 GB (DDR4 2400MHz)
- GPU: NVIDIA GeForce Titan Xp GP102 (Pascal architecture, 3840 CUDA Cores @ 1.6 GHz, 384-bit bus width, 12 GB GDDR G5X memory)

### C.2    Datasets

We describe the details of biased datasets, Colored MNIST (C-MNIST), Corrupted CIFAR-10 (C-CIFAR-10), and BFFHQ.

**Colored MNIST.**    Following existing studies (Nam et al., 2020; Kim et al., 2019; Li and Vasconcelos, 2019; Bahng et al., 2020; Darlow et al., 2020; Lee et al., 2021), this biased dataset comprises two highly correlated attributes: color and digit. We added specific colors to the foreground of each digit, generating bias-aligned and bias-conflicting samples for different ratios of bias-conflicting samples:

- 0.5%: (54751:249)
- 1%: (54509:491)
- 2%: (54014:986)
- 5%: (52551:2449)

**Corrupted CIFAR-10.** Among 15 different corruptions introduced in the original dataset (Hendrycks and Dietterich, 2018), we selected types including Brightness, Contrast, Gaussian Noise, Frost, Elastic Transform, Gaussian Blur, Defocus Blur, Impulse Noise, Saturate, and Pixelate, related to CIFAR-10 classes (Krizhevsky and Hinton, 2009). We used the most severe level of corruption for the dataset, with the following bias-aligned and bias-conflicting samples:

- 0.5%: (44832:228)
- 1%: (44527:442)
- 2%: (44145:887)
- 5%: (42820:2242)

**BFFHQ.** The dataset is created by using the Flickr-Faces-HQ (FFHQ) Dataset (Karras et al., 2019), focusing on age and gender as two strongly correlated attributes. The dataset includes 19200 training images (19104 bias-aligned and 96 bias-conflicting) and 1000 testing samples.

### C.3 Image Preprocessing

Following Lee et al. (2021), our model is trained and evaluated using fixed-size images. For C-MNIST, the size is $28 \times 28$; for C-CIFAR-10, it is $32 \times 32$, and for BFFHQ, it is $224 \times 224$. Images for C-CIFAR-10 and BFFHQ are preprocessed using random crop and horizontal flip transformations, as well as normalization along each channel (3, H, W) with a mean of (0.4914, 0.4822, 0.4465) and standard deviation of (0.2023, 0.1994, 0.2010). We do not use augmentation techniques for C-MNIST.

### C.4 Performance Evaluation

**Training Details.** For training, we use the Adam (Kingma and Ba, 2014) optimizer with default parameters (i.e., betas = (0.9, 0.999) and weight decay = 0.0) provided in the PyTorch™framework. We define two different learning rates: $LR_{DGW}$ for our DGW modules, and LR for the remaining modules in our method, including encoders and classifiers. For C-MNIST, LR is 0.01, while $LR_{DGW}$ is 0.0005 for C-MNIST-2%, 0.002 is for the remaining ratios of datasets. For C-CIFAR-10, LR is 0.001, and $LR_{DGW}$ is 0.0001. For BFFHQ, LR is 0.0001 and 0.0002 is for $LR_{DGW}$.

We utilize StepLR for learning rate scheduling, with a decaying step set to 10K for all datasets. The decay ratio is 0.5 for both C-MNIST and C-CIFAR-10 and 0.1 for BFFHQ. Following (Lee et al., 2021), we adjust the learning rate after performing feature augmentation.

We set the hyperparameters $(\lambda_{re}, \lambda_{swap_b}, \lambda_{swap}, \lambda_{ent})$ for our proposed loss functions (Section 3.3 in the main text). $(10, 10, 1, 0.01)$ is set for the ratio of 0.5% of C-MNIST, and $(15, 15, 1, 0.01)$ for the ratio of 1%, 2%, and 5% of C-MNIST. We set $(1, 1, 1, 0.01)$ for C-CIFAR-10, and $(2, 2, 0.1, 0.01)$ for BFFHQ.

Our proposed mixup strategy uses the hyperparameter $\beta$ to select the mixing coefficient $\alpha \sim \text{Beta}(\beta, \beta)$. For BFFHQ, we set 0.5, whereas 0.2 for C-MNIST and C-CIFAR-10.

We provide the scripts, including all hyperparameter setups, in our Git repository (Section A) to reproduce our performance evaluation.

### C.5 Analysis for Interpretable Attribute Representation

**Initialization of Concept Slots.** The initialization of concept slots is crucial for our model's performance, tailoring the attention mechanisms to each dataset. We set the initial number of concept slots ($C$) as follows:

- For C-MNIST, $C$ is set to 2, reflecting its simple attribute composition
- For C-CIFAR-10, $C$ is set to 10, accommodating its diverse features
- For BFFHQ, $C$ is set to 10, capturing a wide range of human facial features

B

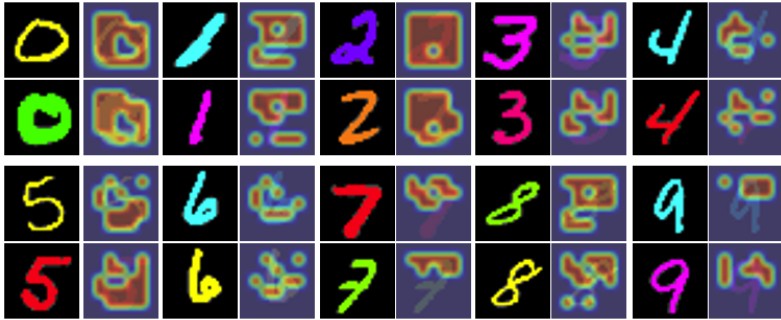

Figure A-1: Visualization of attention masks $\mathbf{A}^i$ for the C-MNIST dataset

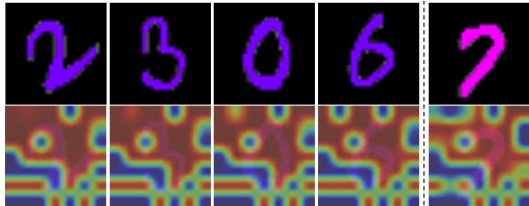

Figure A-2: Visualization from the C-MNIST dataset showing attention masks $\mathbf{A}^b$, highlighting color patterns. Digits in similar colors (e.g., 2, 3, 0, and 6) share similar attention mask patterns.

**Additional Visualization on C-MNIST dataset.** Figure A-1 displays the attention masks $\mathbf{A}^i = \mathbf{A}(\mathbf{S}^i_{\text{latent}}, \mathbf{E}^i)$ generated by eq. 3 in the main text for C-MNIST, showing the model focuses on digit shapes, ignoring color. Fig. A-2 shows the attention masks $\mathbf{A}^b = \mathbf{A}(\mathbf{S}^b_{\text{latent}}, \mathbf{E}^b)$ generated by eq. 3 in the main text, highlighting how the model responds to color patterns. Similar colors, like the purple digits 2, 3, 0, and 6, have similar attention masks, indicating the model's sensitivity to color.

**Visualization on BFFHQ dataset.** Figure A-3 shows DGW's behavior on the BFFHQ dataset, where the intrinsic components display complementary behavior within themselves (concept slots 6 and 9), focusing on specific facial features like cheeks for gender classification. This behavior is due to BFFHQ's focus on human facial shapes for gender classification, where the model prioritizes critical facial features, filtering out less relevant data.

## C.6 Quantitative and Qualitative Analysis

**t-SNE and Clustering.** We provide more results with t-SNE plots and clustering scores with V-Score (Rosenberg and Hirschberg, 2007) as illustrated in Fig. A-4 and A-5. V-Score, a harmonic mean between homogeneity and completeness, is widely used to evaluate clustering. A higher V-Score indicates tighter intra-class clusters and better inter-class separation.

In Fig. A-4, intrinsic features from baselines and the intrinsic attribute encoder $\phi^i$ are used. It consistently shows a higher V-Score, implying better classification and intrinsic attribute capture compared to baselines. V-Scores are higher in setting (ii) than (i) because more bias-conflicting samples are used for training in setting (ii).

In Fig. A-5, features from the bias attribute capturing layer of LFA and the bias attribute encoder $\phi^b$ are utilized. It shows a higher V-Score compared to LFA, indicating more effective bias attribute separation. Overall, our method outperforms baselines, demonstrating robust separation of intrinsic and bias attributes to improve debiasing process.

**Model Similarity.** We use Centered Kernel Alignment (CKA) (Raghu et al., 2021; Kornblith et al., 2019; Cortes et al., 2012) to visualize similarities between all pairs of layers in different models, helping us understand model behavior. The bias and intrinsic attribute encoders $\phi^b$ and $\phi^i$ in our approach are compared.

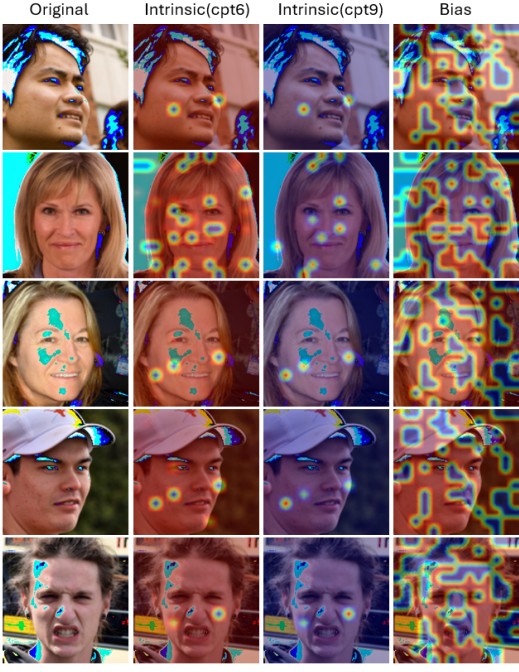

Figure A-3: Face images with attention masks. The first column shows the original image, the next two columns show attention masks $\mathbf{A}^i$ from concept slots 6 and 9, and the last column shows masks $\mathbf{A}^b$.

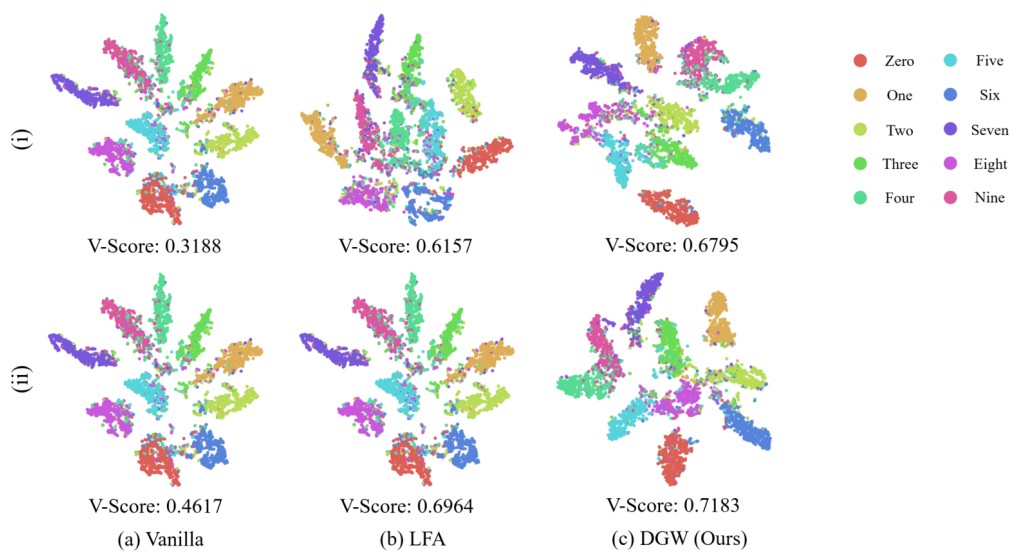

Figure A-4: t-SNE plots for intrinsic features on C-MNIST (with (i) 1.0% and (ii) 2.0% settings).

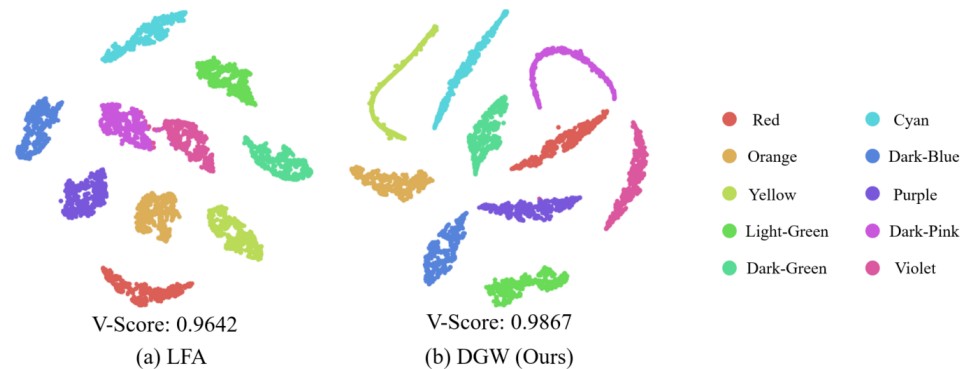

Figure A-5: t-SNE plots for bias features on C-MNIST (with 0.5% setting).

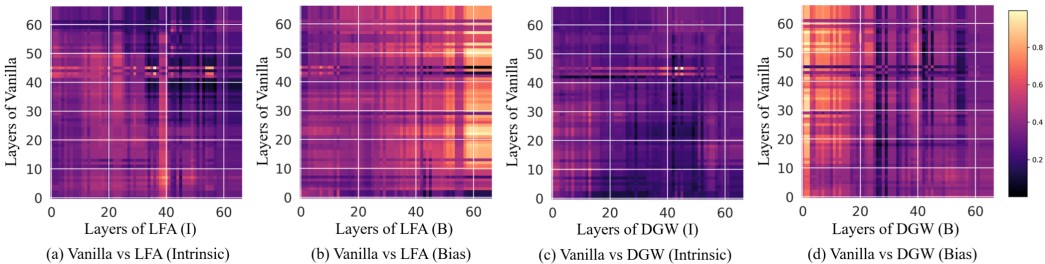

Figure A-6: Representations of similarities for vanilla model and different methods with all pairs of layers on C-CIFAR-10 (5.0% setting). A high similarity score denotes high values.

In Fig. A-6 and Fig. A-7, Vanilla and LFA models show similar weights in many layers, represented by bright colors.

In contrast, our method shows significantly lower similarity values, indicating different weights and behaviors across layers compared to Vanilla and LFA. Our method affects deeper layers more, where the attention module is inserted, suggesting a distinct impact on model behavior.

**Model Reliability.** To evaluate the generalizability of models, we measure Expected Calibration Error (ECE) and Negative Log Likelihood (NLL) (Guo et al., 2017), where ECE is to measure calibration error and NLL is to calculate the probabilistic quality of a model. In detail, ECE aims to evaluate whether the predictions of a model are reliable and accurate, which is a simple yet sufficient metric for assessing model calibration and reflecting model generalizability (Guo et al., 2017).

In Table A-1, our method consistently shows the lowest ECE, indicating better calibration and reliability. For C-MNIST, it presents a higher NLL compared to baselines. Since C-MNIST includes color bias only in the training set, it prevents overfitting by being less affected by bias, leading to better overall model performance. This trend is consistent across different settings in C-MNIST, providing insights into analyzing and explaining dataset bias types and complexity characteristics.

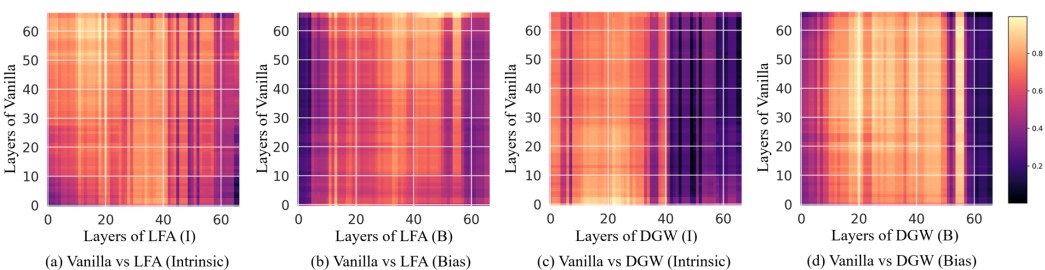

Figure A-7: Representations of similarities for vanilla model and different methods with all pairs of layers on BFFHQ (0.5% setting). A high similarity score denotes high values.

Table A-1: ECE (%) and NLL under different settings on C-MNIST and C-CIFAR-10.

| Dataset | C-MNIST | | | | | | | | C-CIFAR-10 | | | | | | | |
|---|---|---|---|---|---|---|---|---|---|---|---|---|---|---|---|---|
| Ratio (%) | 0.5 | | 1.0 | | 2.0 | | 5.0 | | 0.5 | | 1.0 | | 2.0 | | 5.0 | |
| | ECE | NLL | ECE | NLL | ECE | NLL | ECE | NLL | ECE | NLL | ECE | NLL | ECE | NLL | ECE | NLL |
| Vanilla | 10.9 | **13.17** | 7.97 | **6.45** | 5.70 | **5.71** | 9.54 | 4.10 | 13.75 | 5.99 | 13.14 | 9.87 | 12.25 | 6.65 | 13.76 | 5.99 |
| LFA | 4.35 | 67.72 | 2.79 | 36.46 | 2.09 | 18.35 | 7.59 | **3.09** | 12.09 | 5.81 | **11.45** | 7.27 | 10.25 | 5.14 | 7.56 | 3.09 |
| DGW | **3.41** | 271.71 | **2.03** | 143.36 | **1.73** | 41.44 | **1.61** | 20.19 | **11.85** | **5.71** | 11.53 | **6.88** | **9.96** | **4.41** | **7.55** | **3.01** |

