# OpenReview forum: "A Cognitive Framework for Learning Debiased and Interpretable Representations via Debiasing Global Workspace"
_NeurIPS.cc/2024/Workshop/UniReps — UniReps_

### Official Review · Reviewer_tXY5 · 2024-10-06
**Debiasing framework is thoroughly benchmarked**

**Rating:** 9
**Confidence:** 1

**Review:**

With my limited knowledge in neuroscience; the paper has immense clarity in detailing of the proposed framework. The datasets were appropriately chosen for the benchmarks and analysis for interpretable attribute representation (intrinsic and biased) proves the efficacy of the approach.

The proposed framework is a significant contribution and will find application in multiple NLP and ML tasks.

For readers and practitioners who would like to use this approach, it is advisable to add a section explaining the components of global workspace theory, how it maps to the proposed framework and motivation section to make this framework more intuitive/understood.

---

### Official Review · Reviewer_MBJo · 2024-10-07
**Review for Debiasing Global Workspace: A Cognitive Neural Framework for Learning Debiased and Interpretable Representations**

**Rating:** 6
**Confidence:** 3

**Review:**

This paper proposes a biologically-inspired architecture for what is essentially invariant learning - aiming to learn robust features from potentially biased data. The method is validated empirically on biased versions of the CV MNIST and CIFAR datasets.

Strengths:

The paper is well written, the experiments seem sound and broadly support the claims. I appreciate the visualisations alongside the qualitative evaluation.

Weaknesses:

For all the complexity of the proposed architecture, the results are not very convincing - the performance on the in-distribution data suffers and the improvements on counterfactual examples are marginal.

I am also surprised that other IRM methods are not discussed (e.g. (1)) - they do require partitioning data in environments, but there have been methods proposed to automatically induce those (e.g. (2)).

Some of the claims in the introduction are somewhat detached from reality in my view - e.g. "In modern ML and AI, it has been argued that it is better to build an intelligent system from many interacting specialized modules rather than a single “monolithic” entity to deal with a broad spectrum of conditions and tasks" (ll40-42). I would argue that "modern ML and AI" is where it is precisely because the opposite is the case: large monolithic generalist models that perform well on a variety of tasks and are capable of generalisation to new unseen ones.

(1): https://openaccess.thecvf.com/content/ICCV2021/html/Teney_Unshuffling_Data_for_Improved_Generalization_in_Visual_Question_Answering_ICCV_2021_paper.html

(2): https://arxiv.org/abs/2010.07249

Edit: After finishing the review I have found an existing preprint which has significant overlaps with this work: https://arxiv.org/pdf/2403.14140v1 I recommend the workshop organisers verify that the preprint is indeed from the authors of this submission.

---

### Official Review · Reviewer_JRDL · 2024-10-07
**A new debiasing technique that does not require spurious label annotations**

**Rating:** 6
**Confidence:** 4

**Review:**

This paper presents a novel debiasing framework, called Debiasing Global Workspace (DGW), which is inspired by Global Workspace Theory (GWT) from cognitive science. DGW offers an alternative approach by leveraging GWT. It proposes specialized modules for learning disentangled representations of intrinsic and biased attributes, with a shared "global workspace" to broadcast relevant information to these modules. The DGW architecture has intrinsic and bias attribute specialists and uses attention-based mechanisms to selectively broadcast information. This allows the model to focus on the intrinsic features during backpropagation, ignoring bias features.
The method is tested on several datasets, including Colored MNIST, Corrupted CIFAR-10, and Biased FFHQ. The results show that DGW achieves better or comparable performance to other debiasing methods, with improved interpretability.

**Strengths**

Novelty: The use of Global Workspace Theory (GWT) for debiasing is innovative, especially in how it mimics modular information processing in the human brain.

Interpretability: DGW provides visualizations of attention masks, allowing insights into which features the model relies on for predictions.

Generalizability: DGW does not require predefined bias labels, making it versatile for different datasets and tasks.

**Weaknesses**

Clarity in Explanation:

The introduction of the DGW framework could be more concise. The explanation of how GWT is adapted could benefit from additional clarity, especially for readers unfamiliar with cognitive theories.

 Figure 1 is difficult to read.

Computational Complexity:

While the paper acknowledges that DGW increases training complexity, it would be beneficial to quantify this increase. A direct comparison of training times and computational resources between DGW and baseline models would make the trade-offs clearer.

---

### Official Review · Reviewer_fHHK · 2024-10-07

**Rating:** 4
**Confidence:** 4

**Review:**

The paper combines LfF [49] and LFA [38] with Global Workspace Theory (GWT) (and entropy regularization) into a framework for debiasing called Debiasing Global Workspace (DGW). Specifically, LfF trains two separate classifiers (bias-aligned and bias-conflicting) in an unsupervised manner, exploiting that hard to classify samples are likely bias-conflicting samples. DGW adds an attention module between the representation and classifier part to construct a shared latent space. Additionally, samples are augmented with LFA to increase the amount of bias-conflicting samples.

Strengths:
- S1 The theoretical foundation of GWT is an interesting approach to address the problem
- S2 The approach outperforms unsupervised baselines under consideration in the evaluation

Weaknesses:
- W1 It remains unclear, whether the performance gains are due to DGW or simply due to the combination of LfF and LFA
- W2 While the method is reported as introducing little overhead, the bunch of additional (hyper-)parameters, such as weights for individual loss terms, number of slots and dimensionality in the shared latent space seem to contradict this claim. In particular since different configurations are employed per dataset and ratio of bias-conflicting samples in the training set.
- W3 Whether attention yields interpretability [a1]  or not [a2] in subject to discussion. Either way, the results by anecdotal evidence don't allow to draw strong conclusions, in particular as the interpretation of Fig. 3 is not clear cut.

To me W1 is crucial and I would have expected an ablation study that shows the contribution of individual parts on the performance changes and additionally a baseline that combines LfF and LFA without DGW (e.g., via a simple ensemble). While bibliography is comparably long, related work falls a bit short on recent approaches, such as [a3, a4, a5, ...]. Similarly, an evaluation on commonly used benchmarks in recent work (such as Waterbirds or CelebA) would be desirable.

Minor: the supplementary material indicates that a Titan GPU with 12GB was used for the experiments while footnote 1 indicates that an RTX A6000 with 24GB "could not handle the real-life dataset BFFHQ" (for which reasons?) with SelectMix. Afaik, the A6000 has 48GB, while the older RTX 6000 (without A) has 24GB.

[a1] Attention is not not Explanation - https://arxiv.org/pdf/1908.04626
[a2] Attention is not Explanation - https://arxiv.org/pdf/1902.10186
[a3] Last Layer Re-Training is Sufficient for Robustness to Spurious Correlations - https://arxiv.org/pdf/2204.02937
[a4] Using Early Readouts to Mediate Featural Bias in Distillation - https://arxiv.org/abs/2310.18590
[a5] Towards Last-layer Retraining for Group Robustness with Fewer Annotations - https://arxiv.org/abs/2309.08534
[a6] Deep learning face attributes in the wild - https://openaccess.thecvf.com/content_iccv_2015/html/Liu_Deep_Learning_Face_ICCV_2015_paper.html
[a7] Distributionally Robust Neural Networks - https://openreview.net/pdf?id=ryxGuJrFvS

---

### Decision · Program_Chairs · 2024-10-10

**Decision:**

Accept

**Comment:**

In light of the positive reviewers' feedback and relevancy of the submission, we are pleased to accept this paper for presentation at UniReps 2024. We kindly ask the authors to incorporate the reviewers' suggestions and feedback in the final camera-ready version of the manuscript.